# Reconstructive Approach in Residual Periodontal Pockets with Biofunctionalized Heterografts—A Retrospective Comparison of 12-Month Data from Three Centers

**DOI:** 10.3390/jfb15020039

**Published:** 2024-02-09

**Authors:** Anton Friedmann, Pheline Liedloff, Meizi Eliezer, Arthur Brincat, Thomas Ostermann, Daniel Diehl

**Affiliations:** 1Department of Periodontology, Faculty of Health, Witten/Herdecke University, Alfred-Herrhausen-Str. 50, 58455 Witten, Germany; pheline.liedloff@uni-wh.de (P.L.); daniel.diehl@uni-wh.de (D.D.); 2Independent Researcher, Tel Aviv 3640577, Israel; meizi.eliezer@gmail.com; 3Independent Researcher, 83000 Toulon, France; arthurbrincat@gmail.com; 4Department of Periodontology, Service of Odontology, AP-HM, UFR of Odontology, Aix-Marseille University, 13005 Marseille, France; 5Department of Psychology, Faculty of Health, Witten/Herdecke University, 58455 Witten, Germany; thomas.ostermann@uni-wh.de; 6Institute of Pharmacology and Toxicology, Faculty of Health, Witten/Herdecke University, 58453 Witten, Germany

**Keywords:** heterografts, EMD, xHyA, synthetic polymer barrier, bovine xenograft with hydroxyapatite, allograft

## Abstract

The regenerative capacity of well-preserved blood clots may be enhanced by biologics like enamel matrix derivative (EMD). This retrospective analysis compares outcomes reported by three centers using different heterografts. Center 1 (C1) treated intrabony defects combining cross-linked high-molecular-weight hyaluronic acid (xHyA) with a xenograft; center 2 (C2) used EMD with an allograft combination to graft a residual pocket. Center 3 (C3) combined xHyA with the placement of a resorbable polymer membrane for defect cover. Clinical parameters, BoP reduction, and radiographically observed defect fill at 12-month examination are reported. The 12-month evaluation yielded significant improvements in PPD and CAL at each center (*p* < 0.001, respectively). Analyses of Covariance revealed significant improvements in all parameters, and a significantly greater CAL gain was revealed for C2 vs. C1 (*p* = 0.006). Radiographic defect fill presented significantly higher scores for C2 and C3 vs. C1 (*p* = 0.003 and = 0.014; C2 vs. C3 *p* = 1.00). Gingival recession increased in C1 and C3 (*p* = 1.00), while C2 reported no GR after 12 months (C2:C1 *p* = 0.002; C2:C3 *p* = 0.005). BoP tendency and pocket closure rate shared similar rates. Within the limitations of the study, a data comparison indicated that xHyA showed a similar capacity to enhance the regenerative response, as known for EMD. Radiographic follow-up underlined xHyA’s unique role in new attachment formation.

## 1. Introduction

Periodontitis is a chronic inflammatory disease caused by dysbiotic plaque biofilms that result in irreversible host-mediated damage to the tooth-supporting apparatus [1]. Depending on the rate of disease progression, periodontal bone loss may present itself as vertically configured, so-called intrabony, defects [2]. Regenerative surgery is the preferred method for addressing residual intrabony periodontal defects after non-surgical periodontal therapy, as recommended by the guidelines for treating periodontitis stages 1 to 3 [3]. In a systematic review and meta-analysis, regenerative strategies using enamel matrix derivative (EMD) or guided tissue regeneration (GTR) were found to be more effective than open-flap debridement (OFD) in terms of clinical attachment level (CAL) gain and reduction in probing depth (PD). These strategies showed a significant improvement in CAL gain and PD reduction once implemented [4]. The overall superiority was expressed by a 1.27 mm greater CAL gain achieved with EMD and 1.43 mm achieved with GTR. Furthermore, this systematic review recommended the use of bone substitutes for intrabony defects with severely reduced bone walls to stabilize soft tissue and prevent collapse [2]. Moreover, space maintenance is crucial for both blood clot and tissue formation, according to established GTR principles [5,6]

A meta-analysis calculated the effect named “pocket closure” on behalf of 12 published randomized controlled trials (RCTs) addressing the efficacy of either GTR or EMD over OFD. The results revealed a 61.4% rate of closure looking at sites with ≤3 mm residual probing depth (PD), whereas a 92.1% closure rate was apparent once considering sites with a residual PD ≤ 4 mm after a 12-month post-op period [7].

In recent years, a new agent has proved sufficient in periodontal regeneration after its beneficial role in soft tissue healing had been demonstrated before. Studies conducted in vitro, pre-clinically, and as clinical case series documented the sufficient contribution of adjunctively applied hyaluronic acid to cell and tissue reactions. In particular, cross-linked high-molecular-weight hyaluronic acid (xHyA) showed a sufficient enhancement in soft tissue healing in donor sites for the retrieval of free gingival grafts from the palate [8]. The same formulation effectively supported soft tissue flap stabilization in recession coverage procedures carried out in an animal model as well as in patients [9,10]. The periodontal healing of surgically created intrabony defects was superior regarding a newly formed periodontal ligament and new cementum according to histomorphometric evaluation in a pre-clinical study by Shirakata et al. This group repeated the experiment by creating acute furcation grade 3 defects using the same dog model and confirmed the results from the previous study for the xHyA-treated defects [10,11]. A randomized clinical trial investigating three-wall intrabony defects further demonstrated the non-inferiority of xHyA-treated sites compared to EMD use for surgical regenerative treatment regarding CAL gain and PD reduction outcome after 24 months of follow-up [12]. Moreover, Bozic et al. achieved a >90% pocket closure rate by surgically applying xHyA with a porcine particulate xenograft after 12 months of healing [13].

Apart from clinical studies, the interaction between xHyA and fibroblasts derived from the periodontal ligament was elucidated by an in vitro experiment performed on dentin discs [14]. Another experimental study reported that the presence of xHyA on collagen substrates enhanced the gene transcription rate for bone-related proteins by osteoblast-like cells in vitro [15]. An in vitro study showing xHyA’s impact on the transcription rate of the specific mRNAs encoding for cementoblast differentiation and on their enhanced proliferation was just released [16].

However, while the positive effects of the abovementioned biologics were evident, clinical studies comparing the effects contributed by EMD or xHyA to the regenerative surgical treatment of intrabony periodontal defects still need to be conducted. In this retrospective study, we investigated the outcomes of these two bioactive materials in combination with different adjunctive biomaterials for the surgical regenerative treatment of deep intrabony defects over 12 months.

## 2. Materials and Methods

The patients recruited were routinely treated periodontitis patients presenting with a diagnosis of stage 3 or 4 periodontitis regardless of their grading [17]. In all three centers, patients underwent a course of systematic subgingival instrumentation according to recommendations from the guidelines of the EFP concerning steps 1 and 2 before surgery scheduling [3]. The regenerative approach was favored once the re-evaluation values justified a step-3 surgical therapy of residual pockets, i.e., with a PPD exceeding 6 mm with or without BoP.

Principal investigators represented by A.B. for center 1 (C1), M.E. for center 2 (C2), and A.F. for center 3 (C3) were calibrated regarding the surgical technique applied, data evaluation, and inclusion criteria for approaching the residual pockets. All principal investigators were professionally trained periodontists with similar experience (three-year postgraduate degree, at least 10 years of practicing periodontics), and they performed all the surgeries. The concordant intention was to enhance tissue response to bone substitutes or membranes applied by combining them with bioactive formulations, either enamel matrix derivative (EMD) or high-molecular-weight crosslinked hyaluronic acid (xHyA), aiming at their biofunctionalization. The combination of grafting or membrane material with the bioactive molecules was addressed as a heterograft in each subgroup. Systemically healthy patients were included in this retrospective analysis by each center only. The modified papilla preservation incision design [18], full-flap elevation, and releasing incision for coronal flap advancement, as well as meticulous instrumentation of the root surface and thorough degranulation, were uniformly agreed for the surgical protocol. The protocol standardized neither the type of suture nor the suture technique. There were no restrictions regarding the route and type of instrumentation of the defects; i.e., ultrasonic or piezo devices were used as well as hand instruments. After completing thorough instrumentation, the defects received biomaterials considered supportive for regenerative healing. Each group was free to choose the biomaterial combination for regenerative surgery according to its own preference. The Ethics Committee of Witten/Herdecke University approved the retrospective analysis of the data set from the three centers (S-203/2021, amendment from 2023).

Each operator assessed clinical parameters (PPD, CAL, BoP, and GR for recession) by means of a manual periodontal probe on a regular basis during supportive periodontal therapy (SPT) visits. Clinically assessed values, as well as data regarding defect intrabony defect depth, defect angle, and defect wall number at baseline (prior to surgery) and after a period of 12 months, were reported. The radiographical findings were assessed on periapical 2D radiographs obtained digitally via the parallel technique using a sensor holder (Sidexis, Sirona, Bensheim, Germany) at baseline and 12 months post-op at each center. The calculation of tissue alterations revealed by comparison of both radiographs was reported by each center itself.

Center 1 used a combination of xHyA (HyaDent BG, Regedent AG, Zürich, Switzerland) and a collagen enhanced by hydroxyapatite particles (Collapat II, Symatese, Chaponost, France). Center 2 applied a combination of EMD (Emdogain, Straumann Group, Basel, Switzerland) and an allograft containing 50% cancellous and 50% cortical allograft bone (LifeNet Health, Virginia Beach, VA, USA). Both centers applied the materials to the intrabony pocket closing the site by coronally repositioning the soft tissue flap; neither center used a membrane. For the EMD application, the site was pre-conditioned by using 24% EDTA gel (Pref Gel, Straumann Group), taking care of the bloodless condition of the defect area prior to EMD gel application thereafter, according to recommendations from the manufacturer. The xHyA application also followed the manufacturer’s recommendation; however, any pre-conditioning of the site was redundant and therefore omitted. The rehydration of either bone substitute was carried out on a tray before grafting the defect with a particulate heterograft. The rehydration afforded as much bioactive material as necessary to completely cover the total volume of the graft. The amount of EMD used per site amounted in total to one dose of 0.7 mL, while one ampule of xHyA contained 1.2 mL of the hyaluronic gel. An overview of the applied heterografts is shown in Table 1. 

Center 3 used xHyA alone for filling the intrabony defect component, placing a poly-lactic poly-lactid polymer membrane (Guidor matrix barrier, Sunstar, Schönau, Germany) at the crest of the alveolar ridge before closing the site with a soft tissue flap in a similar way to both other centers. The membrane was rehydrated by xHyA similarly to the rehydration of the bone substitute in the other two centers.

The post-op regimen included pain medication, irrigation with CHX for a duration of 2 weeks, and local topical use of CHX gel for several weeks following suture removal. Each center was responsible for the choice of systemically administrated antibiotics for every case.

For metrical variables, e.g., PPD, CAL, and recession, descriptive statistics including mean, standard deviation, median, range, and percentages were applied to summarize the sample data. Differences between groups were calculated using Analysis of Covariance (ANCOVA) or a chi-square test in the case of nominal data with intraosseous depth, defect angle, and wall number as covariates. For pairwise group comparisons, the Bonferroni post hoc test was used. A two-tailed significance level of α = 5% was applied for all analyses.

## 3. Results

All three centers recorded and reported uncomplicated healing. All patients were compliant with the SPT program and appeared at individual intervals for re-evaluation and cleaning visits. At one-year re-evaluation, all patients from three centers demonstrated significantly improved clinical parameters and positive alterations in crestal bone height when followed up radiographically.

C1 enrolled 18 patients with 19 treated defects, C2 accounted for 21 patients with an equal number of treated teeth, and C3 enrolled 15 patients with 15 teeth and sites to treat, respectively. The homogeneity in patient age and defect morphology included in the three centers was confirmed by non-significant differences in the defect angle, intrabony depth component, number of defect walls, initial probing depth (PD), and clinical attachment loss (CAL) loss. Furthermore, age, gender, and smoking habits were similarly distributed among the patients from each center (Table 2). The correlation between the outcome and the radiographic defect diminution (RDD/∆defect fill) outcome was statistically significant only for the baseline value and the intraosseous defect component (*p* < 0.001); the initial number of bony walls (*p* = 0.174) and the defect angle (*p* = 0.843) were non-significant. Moreover, all groups exhibited a similar distribution of morphologic defect characteristics (Table 2).

While the PD reduction was similarly effective in all three centers (Table 2 and Table 3), the inner group comparison revealed statistically significant differences in attachment-level gain reported by C1, C2, and C3 (*p* < 0.001, respectively) (Table 2; Figure 1). Figure 2, Figure 3 and Figure 4 depict and illustrate one representative case per center including clinical images and periapical X-ray at baseline and 12 months post-op.

The ∆CAL comparison between centers favored center 2 vs. center 1 with a *p* = 0.006; the difference between C2 and C3 was statistically non-significant (*p* = 0.718). The radiographic bone fill was significantly greater in patients from centers 2 and 3 vs. center 1 (*p* = 0.003 and =0.014, respectively) (Table 4). The difference in radiographically documented defect fill between C2 and C3 was statistically non-significant (*p* = 1.0). As corroborated by the 12-month results, both the significant clinical attachment gain and radiographic alveolar bone improvement remained constantly unaltered during the observation period (Table 2, Figure 1, Figure 2 and Figure 3). The recession increased from baseline to the 12-month visit by 1.2–1.3 mm on average for center 1 and 3, while center 2 recorded a minimal recession increase of less than 0.5 mm.

The rate for pocket closure was estimated at >90% in all treated sites regardless of the type of biomaterial. In detail, looking at the residual PD ≤ 4 mm without BoP, C1 showed 89.5%, C2 showed 95.3%, and C3 showed 93.4% pocket closure rates at the level of an residual probing depth of <4 mm without bleeding on probing. BoP appeared sufficiently reduced in all treated defects at an overall rate of 93%, without a great difference between three centers.

Among the covariates, the intraosseous defect depth tested significant for the outcome in ∆PPD, ∆CAL and ∆defect fill (*p* < 0.001). Other covariates, such as defect angle, number of defect walls, localization, or defect width (if reported), were not significantly associated with the clinical outcome (Table 4).

## 4. Discussion

This retrospective data analysis displayed a significant improvement for the patients treated by any of the three centers. Each center achieved a significant reduction in PPD at a clinically relevant level, with pocket closure rates ranging from 90 to 94%. This change was accompanied by significant attachment-level gains in each group. The results are in line with a recent systematic review published by Nibali et al., who reported an adjunctive benefit of regenerative procedures compared with open-flap debridement alone [4]. The authors also emphasized that the addition of deproteinized bovine bone mineral may further improve the outcome of GTR. However, in this retrospective analysis, the clinical attachment gain of C3 was equal to or even superior to the groups that used particulate bone substitutes. This may be rooted in the application of hyaluronic acid, which provides extended blood clot stability and a significant increase in osteogenesis [19,20].

To date, various in vitro studies have validated the beneficial mode of action provided by hyaluronic acid in periodontal tissue regeneration. Via binding to its canonical receptors CD44 and RHAMM, it has been shown to increase proliferation, migration, and cell metabolism in periodontal ligament fibroblasts (PDLs), mesenchymal stromal cells (MSCs), and cementoblasts. In osteoblasts and cementoblasts, xHyA may also stimulate the expression of bone-specific genes, while it shifts macrophage polarization towards an anti-inflammatory M2 phenotype (Figure 5) [14,16,21,22,23,24]. In a recent histomorphometric study in dogs, the combination of a resorbable matrix and xHyA was also shown to be superior, underlining the regenerative capacity of this biomaterial [25]. Moreover, other clinical studies reported that the adjunctive application of hyaluronic acid may improve CAL gain by a significant margin [26].

The regenerative potential of EMD in the formation of new attachment has been confirmed by a plethora of RCTs and human histological as well as in vitro studies (Figure 5) [27,28,29,30,31,32,33]. Quite expectedly, the results of C2 are, therefore, in the range of the latest clinical trials investigating EMD heterografts [34,35]. However, a comparison of our study results with the literature should be made with reasonable caution since the presented study was neither randomized nor controlled.

Nevertheless, the effectiveness of the chosen material composition proved to result in statistically significant differences from the baseline outcomes each center reported. Looking at the baseline number of bony walls, which were almost alike in all three centers (*p* = 0.137), the results from center 3 reported for the first time a significant CAL and bone gain accompanying xHyA use without a bone substitute in defects presenting a diminished number of bone walls (i.e., 1.5 on average). Stabilizing the defect via a polymer-derived membrane, the pocket closure effect at the level of a residual 3 mm probing depth was constantly observed after 12 months (93%).

The EFP guidelines recommend that in the case of a minimized number of bony walls, combined use with a particulate bone graft prevents the risk of tissue collapse into the defect. Thus, centers 1 and 2 combined the use of biologics (xHyA, EMD) with either a xenograft (C1) or an allograft (C2) according to the recommendations for treating defects characterized by a diminished number of walls [36,37]. Since C3 yielded significantly higher CAL gain values, the inconsistency with C1 may have rather been rooted in the choice of the substitute material. The bone substitute was a fleece with 98.9% porosity that has been proposed for tissue engineering [38]. The xenograft was recommended as a hemostatic device for enclosed defects [39]. Used in this series as a graft in an open periodontal pocket environment, even in combination with xHyA, the material may have undergone more rapid degradation compared with the allograft used by C2. The high proportion of collagen in this biomaterial may have been responsible for a rapid resorption accompanied by a partial collapse of the flap into the intraosseous defect. This healing pattern was then associated with an increasing post-operative recession and reduced the potential for the regeneration of the intraosseous component. Previous studies have pointed out that the type of collagen may be crucial for supporting the bone regeneration process and fast collagen degradation may be associated with limited outcomes [40].

Center 3, on the other hand, did not use any particulate material to avoid an artificial radio-opacity in the defect area and applied a polymer barrier membrane instead. This polymer membrane was shown to significantly improve clinical attachment levels in a variety of clinical studies [41]. In a 6-year observational study, Stavropoulos and Karring showed an attachment-level gain with a PPD improvement of 3.8 ± 1.1 mm and a mean CAL gain of 3.8 ± 1.4 mm observed after 1 year. Within this frame of reference, it appears reasonable to suggest that the inconsistency between C1 and C3 was indeed rooted in the choice of substitute material and the fact that xHya was demonstrated to be an efficient addition to the GTR technique applied by center 3 [42].

From this standpoint, it is rational to suggest that more research should focus on the substitute material within a functionalized heterograft, as EMD and xHya already exhibit sufficient evidence. Unfortunately, the membrane material used by C3 in combination with xHyA has been withdrawn from the market by the manufacturer due to recent legacy MDR regulations for European countries. Thus, making a warranted reproduction in a prospective study design impossible for the moment. Nevertheless, further studies comparing heterografts and guided tissue regeneration with the same biological agent would be meaningful.

All patients treated in this retrospective series were systemically healthy to exclude potential confounders. Nevertheless, the centers were not calibrated in terms of post-operative pain medication or antibiotics. As indicated by Table 2, different analgesic medications were applied throughout the healing period if required. Non-steroidal anti-inflammatory drugs (NSAIDs) have been proposed as an adjunct to periodontal treatment repeatedly, for obvious reasons [43]. Being cyclooxygenase inhibitors, they have been shown to reduce periodontal inflammation substantially [44,45]. However, adjunctive effects of reasonable dosing schemes on periodontal therapy were not confirmed by clinical studies [46]. Bearing in mind that the actual usage of NSAIDs was even less prominent in this study, it appears quite unlikely that the analgesic medication may pose a source of bias in this analysis. 

Regarding antibiotics, a recent systematic review and meta-analysis evaluated the possible adjunctive effects of antimicrobial drugs on periodontal regenerative surgery. From a total of 105 randomized clinical trials, the authors were unable to detect any additional benefits to the treatments in terms of CAL gain, indicating that the varying antibiotics prescribed in this study had no significant impact on the clinical or radiographic outcomes [47].

Taken together, all heterografts yielded significant improvements in a range that is expected from the clinical literature. However, owing to the retrospective nature, this study has some limitations. Since no randomization was applied and no controls were included, intergroup comparisons should be regarded with caution. However, one must bear in mind that extensive trials comparing both EMD and xHya in a randomized setting are not available. Moreover, the clinical literature provides sufficient evidence for the regenerative efficacy of both biomaterials, rendering resource-intensive clinical studies for the purpose of identifying a superior biologic at least questionable. Also, measurable differences would possibly just be found regarding the handling of the materials in a heterograft and not the clinical outcome, since EMD usually requires defect surfaces free of blood. This, however, remains a matter of speculation. 

In all three centers, as well as in the relevant literature, the CAL gain and defect fill were subject to quite high standard deviations. While this is a natural occurrence in clinical studies that can be dealt with by an adequately powered study design, it remains intriguing to understand whether patients exhibited interindividual differences in susceptibility for a specific biomaterial. For instance, the HA-binding receptor CD44 naturally expresses differing transcript variants, one of which lacks HA-binding ability [27,48]. An overexpression of this specific variant in the periodontal tissues may thus lead to a diminished efficacy of xHya-functionalized grafts. Identifying individual markers for biomaterial susceptibility may provide clinicians with an evidence-based toolkit for targeted regenerative therapies in the future.

## 5. Conclusions

In summary, this retrospective analysis of biofunctionalized heterografts across three centers demonstrated significant improvements in pocket-probing depth reduction and attachment-level gains. Center 3 showcased noteworthy clinical attachment-level improvements without using particulate bone substitutes, emphasizing the potential benefits of hyaluronic acid application in GTR. While the study’s limitations are acknowledged, the findings highlight the effectiveness of biologics like EMD and hyaluronic acid in periodontal regenerative surgery, prompting the need for future randomized controlled trials to optimize treatment combinations for enhanced patient outcomes.

## Figures and Tables

**Figure 1 jfb-15-00039-f001:**
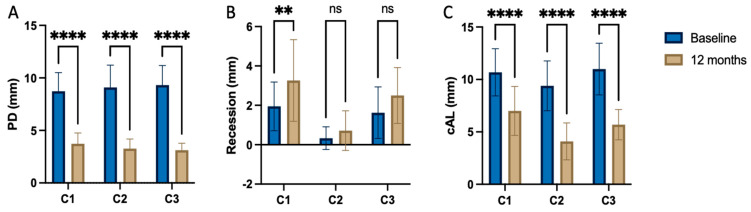
Intragroup comparisons between preoperative and post-operative clinical parameters (**A**) probing depth, (**B**) Recession and (**C**) clinical attachment loss. ** *p* < 0.01; **** *p* < 0.0001; ns—non-significant.

**Figure 2 jfb-15-00039-f002:**
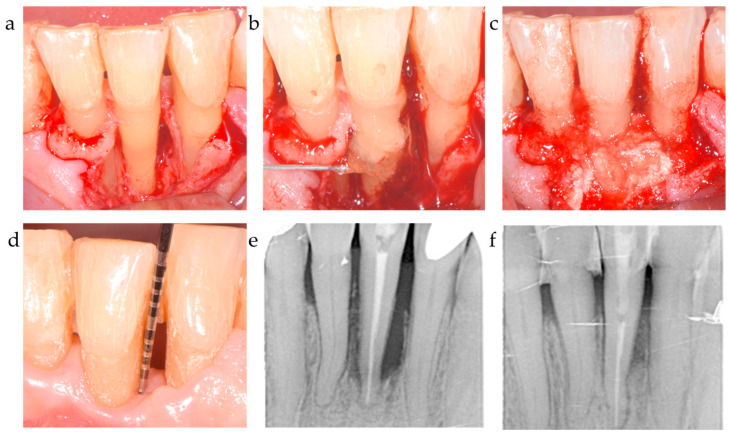
Center 1 showcase: (**a**,**b**) radiographically observed change in bone level around tooth 31 before surgery and 12 months post-op; (**c**,**d**) defect extension and defect grafting at surgery; (**e**,**f**) the result of grafting.

**Figure 3 jfb-15-00039-f003:**
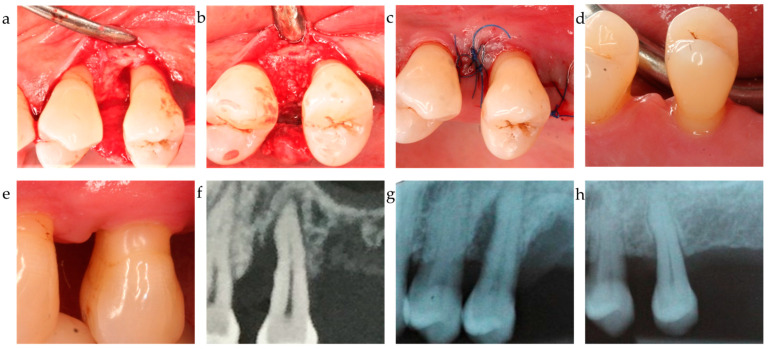
Center 2 showcase: (**a**–**c**) defect extension, defect grafting at surgery, and suture; (**d**,**e**) clinical outcome at 12-month exam. (**f**–**h**) radiographically observed change in bone level around tooth 25 before surgery and 12 months post-op.

**Figure 4 jfb-15-00039-f004:**
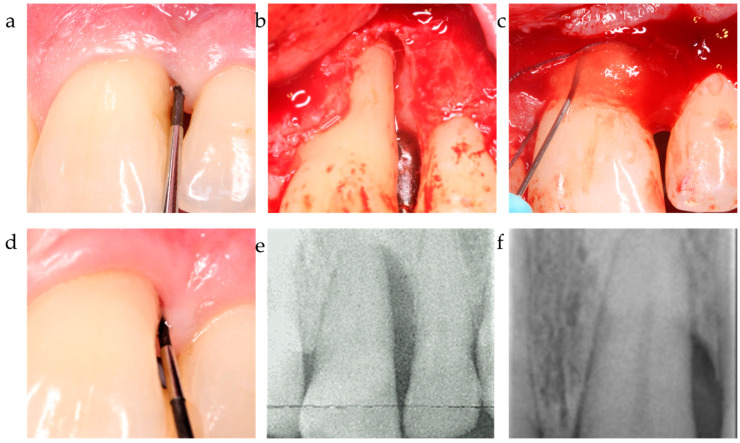
Center 3 showcase: (**a**,**b**) PPD and defect extension; (**c**,**d**) defect grafting at surgery as result of barrier placement and xHya application, and clinical outcome at 12-month exam. (**e**,**f**) radiographically observed change in bone level around tooth 21 before surgery and 12 months post-op.

**Figure 5 jfb-15-00039-f005:**
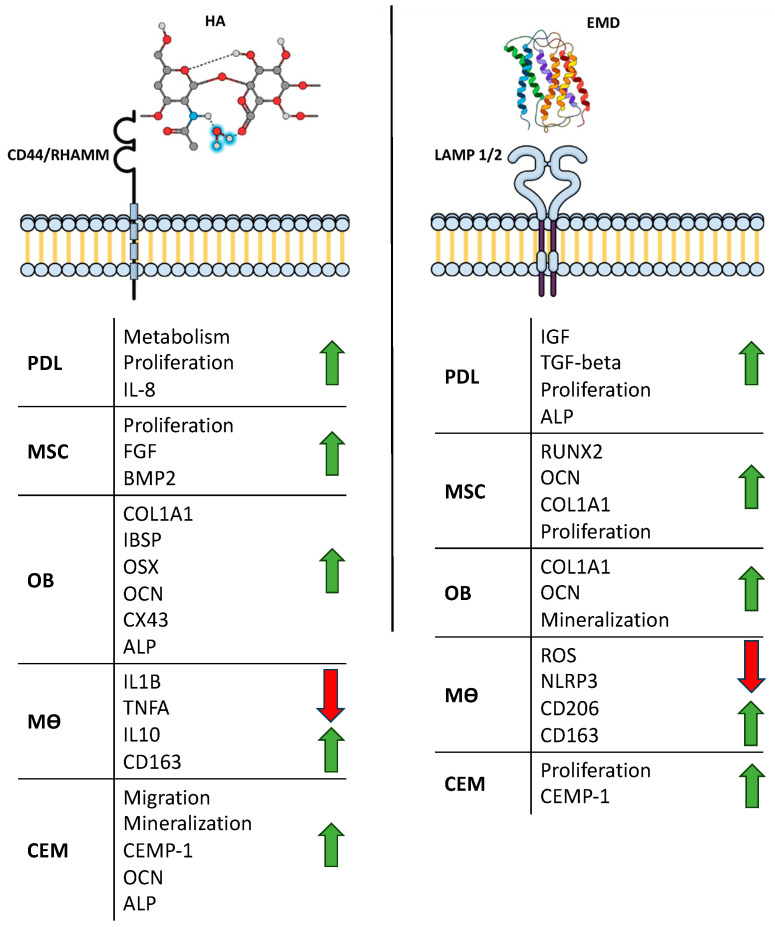
Schematic illustration of the mechanism of action reported for xHya (left) and EMD (right). PDL = periodontal ligament fibroblast; MSC = mesenchymal stromal cell; OB = osteoblast; MO = macrophage; CEM = cementoblast. Red arrows indicate downregulation, green arrows indicate upregulation.

**Table 1 jfb-15-00039-t001:** Heterografts used in the study.

**C1**	xHyA	BDDE-crosslinked hyaluronic acid	HyaDent BG, Regedent, Zürich, Switzerland
	Collapat II	Bovine collagen + dispersed hydroxyapatite granules	Collapat II, Symatese, France
**C2**	EMD	Enamel matrix derivative, Propylenglycolalginate (PGA), water	Emdogain, Straumann Group, USA
	OraGraft	Cortical/cancellous mineralized particulate 50/50	LifeNet Health, USA
**C3**	xHyA	BDDE-crosslinked hyaluronic acid	HyaDent BG, Regedent, Zürich, Switzerland
	Guidor matrix barrier	Polylactic polymer	Sunstar, Germany

**Table 2 jfb-15-00039-t002:** Patient demographics, pooled defect characteristics, and allocation per center (C1–C3).

	C1 (n = 19)	C2 (n = 21)	C3 (n = 16)	Total (n = 56)	*p*-Value
**Age (years)**					0.004
Mean ± SD	58.5 ± 9.2	46.6 ± 9.3	53.1 ± 13.9	52.5 ± 11.7	
Median	57	46	54	55	
Min	36	32	20	20	
Max	75	65	75	75	
**Gender**					0.085
Male	10 (52.6%)	4 (19.0%)	6 (18.75%)	17 (30.4%)	
Female	9 (47.4%)	17 (81.0%)	10 (62.5%)	36 (64.3%)	
**Smoker**					0.192
Yes	3 (15.8%)	4 (19.0%)	0 (0.0%)	7 (12.5%)	
No	16 (84.2%)	17 (81.0%)	16 (100.0%)	49 (87.5%)	
**Localization**					0.480
Mandible	13 (68.4%)	11 (52.4%)	11 (68.8%)	35 (62.5%)	
Maxilla	6 (9.47%)	10 (9.02%)	5 (9.22%)	21 (37.5%)	
**Walls (n=)**					0.137
1	4 (21.1%)	1 (4.8%)	6 (37.5%)	11 (19.6%)	
2	11 (57.8%)	17 (81.0%)	8 (50.0%)	36 (64.3%)	
3	4 (21.1%)	3 (14.3%)	2 (12.5%)	9 (16.1%)	
**Intraosseous depth (mm)**					0.210
Mean ± SD	5.96 ± 1.68	5.70 ± 3.50	7.23 ± 2.44	6.22 ± 2.73	
Median	5.9	5.1	7.8	6.0	
Min	3.3	1.9	2.4	1.9	
Max	10.6	15.3	11.0	15.3	
**Defect angle (°)**					0.508
Mean ± SD	28.54 ± 9.80	31.32 ± 9.66	32.88 ± 14.22	30.82 ± 11.12	
Median	29.6	32.4	31.35	31.7	
Min	14.0	14.6	16.5	14.0	
Max	50.5	44.2	64.5	64.5	
**Defect width (mm)**					0.375
Mean ± SD	2.81 ± 099	2.55 ± 0.86	N/A	2.62 ± 0.92	
Median	2.7	2.4		2.55	
Min	1.5	1.1		1.1	
Max	4.8	4.5		4.8	
**Antibiotics**					
Duration (n)	7 days (19)	--	10 days (16)		
Type (mg)	Amoxicillin (2000)	--	Doxycycline (200)		
**Analgesics**					
Duration (n)	If required	If required	If required		
Type (mg)	Prednisone + Paracetamol (80 + 1000)	Ibuprofen (400)	Ibuprofen (600)		

**Table 3 jfb-15-00039-t003:** Center-allocated change between baseline and 12-month exam for clinical parameters.

	C1		C2		C3	
	Baseline	12-mo	Baseline	12-mo	Baseline	12-mo
**PPD (mm)**						
Mean ± SD	8.74 ± 1.82	3.74 ± 1.05	9.29 ± 2.13	3.38 ± 0.92	9.50 ± 1.86	3.19 ± 0.66
Median	8.00	4.00	9.00	3.00	9.50	3.00
Minimum	7	2	7	2	6	2
Maximum	13	6	12	6	12	4
*p*-value	<0.001		<0.001		<0.001	
**CAL (mm)**						
Mean ± SD	10.68 ± 2.31	7.00 ± 2.29	9.62 ± 2.38	4.10 ± 1.76	11.25 ± 2.46	5.69 ± 1.45
Median	10.00	7.00	9.00	4.00	11.00	6.00
Minimum	7	4	7	2	7	3
Maximum	16	15	16	9	15	8
*p*-value	<0.001		<0.001		<0.001	
**REC (mm)**						
Mean ± SD	1.95 ± 1.27	3.26 ± 2.13	0.33 ± 0.58	0.71 ± 1.10	1.62 ± 1.31	2.50 ± 1.41
Median	2.00	3.00	0.00	0.00	1.00	3.00
Minimum	0	0	0	0	0	0
Maximum	5	10	2	3	4	5
*p*-value	0.003		0.008		0.029	

**Table 4 jfb-15-00039-t004:** Comparison of Δ-values between centers and covariate analyses.

	C1 (n = 19)	C2 (n = 21)	C3 (n = 16)	*p*-Value				Significant Covariates
				Overall	C1 vs. C2	C2 vs. C3	C1 vs. C3	
**ΔPPD**	4.95 ± 1.71	5.81 ± 1.78	6.25 ± 1.88	0.192	0.287	1.00	0.476	Intraosseous depth (*p* < 0.001)
**ΔCAL**	3.68 ± 1.67	5.86 ± 2.37	5.53 ± 1.92	0.007	0.006	0.718	0.158	Intraosseous depth (*p* < 0.001)
**REC**	1.32 ± 1.67	0.04± 0.01	1.33 ± 1.11	0.015	0.031	0.038	1.00	-
**Δdefect fill**	3.33 ± 1.76	4.95 ± 2.43	5.97 ± 2.49	0.002	0.003	1.00	0.014	Intraosseous depth (*p* < 0.001)

## Data Availability

The data that support the findings of this study are available upon reasonable request from the corresponding author, A.F. The data are not publicly available due to ethical concerns regarding patient information.

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
