# Peer review of "Reconstructive Approach in Residual Periodontal Pockets with Biofunctionalized Heterografts—A Retrospective Comparison of 12-Month Data from Three Centers"

_jfb, 2024, doi:10.3390/jfb15020039_

Round 1
Reviewer 1 Report
Comments and Suggestions for Authors
The authors are requested to add a table for the materials used in the current study describing in details its contents.
The authors are requested to add a schematic illustration in the discussion section describing the mechanism of action of the tested materials.
Comments on the Quality of English LanguageThe English language does not has major problems
Author Response
Reviewer 1
The authors are requested to add a table for the materials used in the current study describing in details its contents.
Agreed. We have added a new Table 1 to the method section including the contents.
The authors are requested to add a schematic illustration in the discussion section describing the mechanism of action of the tested materials.
We have added Figure 5 in the manuscript, which summarizes the mechanism of Action of both materials.
Reviewer 2 Report
Comments and Suggestions for Authors
In introduction aspects about etiology and pathogenesis of periodontal diseases.
Explain if antibioterapy influences the results?
pain medication ? is antiinflammatory or analgesic ?
Posible table summarizing the results from all three centers.
Author Response
Reviewer 2
In introduction aspects about etiology and pathogenesis of periodontal diseases.
Agreed. We have amended an introductory sentence (line 36-37).
Explain if antibioterapy influences the results?
Thank you for pointing this out. The type and the amount of post-op medication are now listed in Table 1. As we have highlighted throughout the manuscript, the three centers chose their own treatment algorithms, except for the surgical technique. For that reason, C1 has applied systemic antibiotics and the other two did not. However, we have added a small paragraph to the discussion that picks up on the issue of intergroup comparability. Please refer to the lines 337-350.
pain medication ? is antiinflammatory or analgesic ?
In all three centers different postoperational pharmaceutic regimens were applied. We have added the exact analgesics to Table 1. In all centers, the analgesics were both anti-inflammatory and analgesic. However, as we now discuss in lines 337-350, the use was restricted to individual needs for post-op pain. To date, evidence suggests that anti-inflammatory or antibiotic medication has no measurable effect on regenerative outcomes, which is why we have not prohibited the prescription of one’s favorite drug by protocol. Moreover, there are differences in availability for certain analgesics or antibiotics between the countries of residency of the centers. The total amount of analgesics used by patients from each center was much too low to expect any regenerative or just even an anti-inflammatory effect. The amount of NSAIDs required to initiate clinical effects accordingly has been shown to be much greater then was administrated in this study.
Posible table summarizing the results from all three centers.
We kindly disagree with the reviewer. We would like to point out that Table 1 includes all data from the three different centers.
Reviewer 3 Report
Comments and Suggestions for Authors
Dear Authors,
The manuscript entitled "Reconstructive approach in residual periodontal pockets with biofunctionalized heterografts - a retrospective comparison of 12-month data from three centers" is very interesting and relevant in the field of biodental materials used in periodontology. The manuscript compares the 12-month evaluation reported by 3 centers using different heterografts (C1: cross-linked heavy-weight hyaluronic acid (xHyA) with a xenograft used on 18 patients; C2: EMD with an allograft 20 combination to graft residual pocket applied on 21 patients; and C3: combined xHyA with the placement of a resorbable polymer membrane for defect cover in 15 patients).
The manuscript addresses a specific treatment in the periodontology field. In this research, new, specific types of dental biomaterials were used in the treatment of 3rd and 4th class periodontal pockets. The manuscript carries out their comparative study in three research centers.
The authors presented the aims of their research.
The manuscript presents scientific sound and argues the importance of the research.
The introduction section presents the studies that were previously published (15) regarding the treatment of 3rd and 4th degree periodontal disease, and all citations are in accordance with current research.
The Ethics Committee of Witten/Herdecke University (S-203/2021, amendment from 2023) approved the retrospective comparison of 12-month data from the three centers.
Materials and methods are clearly presented, including the characteristics of the studied groups and the statistical analysis used.
The analyzed variables are presented, and the statistical analysis is realized.
The results section presents the baseline evaluation and the one-year reevaluation.
All patients treated in the three centers presented uncomplicated healing, were compliant with the evaluation program, and demonstrated improved clinical parameters. Follow-up assessments were realized clinically and radiographically.
All presented tables and figures are clear, suggestive, and appropriate.
The presented authors demonstrated the beneficial effect of applied heterografts, and the data support all the conclusions drawn by the authors. Conclusions are clearly presented and respond to the main question posed.
Cited references are relevant, but of a total of 30 cited articles, 12 references (40%) are older than 5 years.
The authors should clarify what it means at the end of Table 2: ”* Tables may have a footer” ? (Line 210).
According to my opinion, what is presented between lines 216–243 belongs to the chapter on results. In the discussion chapter, the results of the study should be presented in comparison with those of other authors who have researched this topic.
It should be specified if analgesic and/or anti-inflammatory medication was indicated and in what quantity (in materials and methods), and if it was necessary to be used by the patients (in results).
A table summarizing the results from all three centers would visualize the obtained results much more clearly.
In future research, additional controls over a longer period of time should be considered.
The clinical relevance of the study should be underlined before the conclusions' section.
Authors should be careful with text editing (correctness of the words in the text, spaces, etc.).
For the reasons given above, I consider that the manuscript deserves to be published, but after making minor corrections, regarding the previously mentioned observations.
Author Response
Reviewer 3
Dear Authors,
The manuscript entitled "Reconstructive approach in residual periodontal pockets with biofunctionalized heterografts - a retrospective comparison of 12-month data from three centers" is very interesting and relevant in the field of biodental materials used in periodontology. The manuscript compares the 12-month evaluation reported by 3 centers using different heterografts (C1: cross-linked heavy-weight hyaluronic acid (xHyA) with a xenograft used on 18 patients; C2: EMD with an allograft 20 combination to graft residual pocket applied on 21 patients; and C3: combined xHyA with the placement of a resorbable polymer membrane for defect cover in 15 patients).
The manuscript addresses a specific treatment in the periodontology field. In this research, new, specific types of dental biomaterials were used in the treatment of 3rd and 4th class periodontal pockets. The manuscript carries out their comparative study in three research centers.
The authors presented the aims of their research.
The manuscript presents scientific sound and argues the importance of the research.
The introduction section presents the studies that were previously published (15) regarding the treatment of 3rd and 4th degree periodontal disease, and all citations are in accordance with current research.
The Ethics Committee of Witten/Herdecke University (S-203/2021, amendment from 2023) approved the retrospective comparison of 12-month data from the three centers.
Materials and methods are clearly presented, including the characteristics of the studied groups and the statistical analysis used.
The analyzed variables are presented, and the statistical analysis is realized.
The results section presents the baseline evaluation and the one-year reevaluation.
All patients treated in the three centers presented uncomplicated healing, were compliant with the evaluation program, and demonstrated improved clinical parameters. Follow-up assessments were realized clinically and radiographically.
All presented tables and figures are clear, suggestive, and appropriate.
The presented authors demonstrated the beneficial effect of applied heterografts, and the data support all the conclusions drawn by the authors. Conclusions are clearly presented and respond to the main question posed.
Thank you, the authors appreciate the reviewer’s thorough appraisal.
Cited references are relevant, but of a total of 30 cited articles, 12 references (40%) are older than 5 years.
We partly agree with the reviewer’s assessment, as we believe that 5-year-old publications may still be relevant and state-of-the-art, provided the results have not been refuted in the meantime. Nevertheless, we have added more actual references to the manuscript.
The authors should clarify what it means at the end of Table 2: ”* Tables may have a footer” ? (Line 210).
The phrase was removed, as it was placed simply as a space holder for possible asterisks, thank you.
According to my opinion, what is presented between lines 216–243 belongs to the chapter on results. In the discussion chapter, the results of the study should be presented in comparison with those of other authors who have researched this topic.
Thank you for pointing this out. We adapted the text of the Results section accordingly, however, avoiding redundancy in statements by omitting a passage you are addressing at Discussion section. We have also amended some more references to the relevant literature and discussed our results critically. We hope that this amendment is sufficient for the reviewer.
It should be specified if analgesic and/or anti-inflammatory medication was indicated and in what quantity (in materials and methods), and if it was necessary to be used by the patients (in results).
Thank you for pointing this out. We have addressed this issue in Table 1, which now includes the quantities and durations for antibiotics and analgesics, as well as the number of patients that received either. Moreover, we discuss the use of anti-inflammatory and antibiotic drugs in 337-350
A table summarizing the results from all three centers would visualize the obtained results much more clearly.
The authors would like to point to Tables 2 and 3, that summarize all clinical variables from all centers.
In future research, additional controls over a longer period of time should be considered.
The authors appreciate the recommendation and plan to collect the date accordingly in the future.
The clinical relevance of the study should be underlined before the conclusions' section.
Agreed. We have added the clinical relevance of the study to the discussion in line 377.
Authors should be careful with text editing (correctness of the words in the text, spaces, etc.).
We thoroughly checked the manuscript for typos and spelling now considering it valid in terms of editing.
For the reasons given above, I consider that the manuscript deserves to be published, but after making minor corrections, regarding the previously mentioned observations.
Reviewer 4 Report
Comments and Suggestions for Authors
Friedmann et al “Reconstructive approach in residual periodontal pockets with biofunctionalized heterografts- a retrospective comparison of 12-months data from three centres”
Dear Authors,
Please answer the following questions:
1. ..stage 3 and 4 patients were included regardless of their grading. In this case, please describe, that besides smoking (in case of which dose dependence is also an issue) what systemic diseases these patients had! E.g.: metabolic diseases alone can have a major effect on regeneration. (Confounder effect-limitation of study)
2. In periodontal therapy the practice of the periodontist has a substantial positive effect on the therapeutical outcome. Please, give the data on how many periodontists have taken part in the surgical treatment of these patients in the centres, and how many years of practice they had!
3. Were the partaker doctors calibrated to assure the comparability of the initial and further measurements? This may question comparability of centres.
4. Deeper PPD can reduce more, and more CAL can be gained if recession is not that significant. (Ramfjord et al (1975), Knowles et al (1979), until Werner et al (2023))
5. RCT: where it is mentioned first, please give the whole expression (line 50)
6. Practice years and systemic disease may accumulate a significant effect on the CAL gain of regenerative periodontal surgeries. Including these factors to the statistical evaluation can show what was the primary: humans or used material!
Author Response
Reviewer 4
Dear Authors,
Please answer the following questions:
- ..stage 3 and 4 patients were included regardless of their grading. In this case, please describe, that besides smoking (in case of which dose dependence is also an issue) what systemic diseases these patients had! E.g.: metabolic diseases alone can have a major effect on regeneration. (Confounder effect-limitation of study)Please consider the statement we added to M&M section regarding the systemic condition of the patients:
“Systemically healthy patients were included into this retrospective analysis by each center only.”
- In periodontal therapy the practice of the periodontist has a substantial positive effect on the therapeutical outcome. Please, give the data on how many periodontists have taken part in the surgical treatment of these patients in the centres, and how many years of practice they had!
We completely agree with the reviewer, that training may be a possible confounder regarding intergroup comparisons. All investigators were equally well trained, and we have pointed this out in detail in lines 99-100.
- Were the partaker doctors calibrated to assure the comparability of the initial and further measurements? This may question comparability of centres.
Since no randomization allocated treatment options within each center, this poses a slight limitation to our study. However, we believe that this is discussed adequately in the last paragraph of the discussion. Moreover, all operators were adequately calibrated regarding the measurements, as given in line 96.
- Deeper PPD can reduce more, and more CAL can be gained if recession is not that significant. (Ramfjord et al (1975), Knowles et al (1979), until Werner et al (2023))
Thank you for this comment. Although the statement regarding the relationship between sites presenting with deeper PPD and their greater potential to gain CAL has been confirmed by several studies, it does not apply to this analysis. Please take into consideration the baseline values for PPD and CAL loss indicated by Table 2. The CAL loss, which exceeded the PPD value at baseline in all three centres, indicates that the recession of gingival margin occurred already previously to the planned surgical intervention. Such observations agree with the tissue response to initial subgingival instrumentation of deep pocket sites prior to surgery. Accordingly, the additional extension of recession within the 12-month post-op period is considered likely moderate for all centres involved and did not decline the effect of CAL gain documented for all three treatment options.
- RCT: where it is mentioned first, please give the whole expression (line 50)
Done.
- Practice years and systemic disease may accumulate a significant effect on the CAL gain of regenerative periodontal surgeries. Including these factors to the statistical evaluation can show what was the primary: humans or used material!
Thank you for pointing to this important topic. However, in our case, we had three centers and three dentists, i.e., one per center. Thus, from a statistical point of view, the practical experience is already included in the Center variable. Since all patients were systemically healthy we now addressed the topic of analgesics and antimicrobials more comprehensively in the discussion. We hope this is a reasonable amendment for the reviewer.
Reviewer 5 Report
Comments and Suggestions for Authors
Some major points to consider:
1. The methodology lacks clarity and precision, making it challenging to replicate the study across different centers. Ambiguities in the application of EMD and xHyA could significantly impact the study's reliability.
2. There are inconsistencies in the reporting of data across different centers, raising concerns about the reliability of the results. Standardized reporting is crucial for the validity of a multi-center study.
3. Limited Contextualization of Findings: The manuscript falls short in providing a comprehensive discussion of the broader implications and limitations of the study. Without a clear understanding of the study's context, its significance for the field remains uncertain.
4. The literature review lacks a robust comparison of the study's findings with existing research on EMD and xHyA. Without a thorough integration with the current body of knowledge, the study's novelty and relevance are diminished.
5. The manuscript does not adequately address potential biases in the study design, implementation, and analysis. A lack of consideration for biases could undermine the internal validity of the study and compromise the credibility of the results.
Consider the following points to improve the quality of the manuscript:
1. The term "biofunctionalized heterografts" is introduced but not explicitly defined. Consider providing a clear definition early in the introduction to aid reader comprehension.
2. Specify the specific regenerative mechanisms targeted by enamel matrix derivative (EMD) and cross-linked heavy molecular weight hyaluronic acid (xHyA) in the context of periodontal regeneration.
3. Include studies that investigate the specific cellular and molecular mechanisms through which EMD and xHyA contribute to periodontal tissue regeneration.
4. Clarify the rationale for selecting EMD and xHyA as biofunctionalized heterografts, highlighting their distinct properties and potential synergies.
5. Provide more information on the standardization process for applying EMD and xHyA across different centers.
6. Clearly define the criteria used to classify a site as a success or failure, particularly in terms of pocket closure rates, to ensure consistency across centers.
7. Specify if there were any noticeable variations in the outcomes among different centers, and discuss potential reasons for such variations.
8. Emphasize the specific advantages and disadvantages of using EMD and xHyA as biofunctionalized heterografts for periodontal regeneration.
9. Discuss the practical implications of the study's findings in the context of developing targeted regenerative therapies.
10. Summarize the unique contributions of the study to the understanding of periodontal regeneration using EMD and xHyA.
11. Include figures or tables that visually compare the regenerative outcomes of EMD and xHyA across different centers.
12. Ensure that recent and relevant studies exploring the regenerative potential of EMD and xHyA in periodontal tissues are included in the literature review.
13. Discuss the potential translation of the study findings into clinical practice, addressing any challenges or considerations.
14. Propose specific avenues for future research, such as investigating the long-term effects of EMD and xHyA in diverse patient populations.
Comments on the Quality of English LanguageMinor check is required.
Author Response
Reviewer 5
Some major points to consider:
- The methodology lacks clarity and precision, making it challenging to replicate the study across different centers. Ambiguities in the application of EMD and xHyA could significantly impact the study's reliability.
Please see our comment below; it addresses the description of how both bioactive materials were applied at all centers. Please refer to point 5 at the bottom of our response.
- There are inconsistencies in the reporting of data across different centers, raising concerns about the reliability of the results. Standardized reporting is crucial for the validity of a multi-center study.
We kindly disagree with the reviewer’s assessment of reporting inconsistencies. While the three centers have indeed applied different treatment regimens to their patients, we believe we took care to be as consistent as possible in our measurements and data handling. All three centers have been calibrated in terms of follow-up visits and data to collect. Therefore, we kindly ask the reviewer to give us an example of this so we can improve the quality of the manuscript.
- Limited Contextualization of Findings: The manuscript falls short in providing a comprehensive discussion of the broader implications and limitations of the study. Without a clear understanding of the study's context, its significance for the field remains uncertain.
Agreed. We have rewritten our discussion to provide a better insight to the field and contextualize our results.
- The literature review lacks a robust comparison of the study's findings with existing research on EMD and xHyA. Without a thorough integration with the current body of knowledge, the study's novelty and relevance are diminished.
To the best of our knowledge, there are just a few studies available that report on differences between outcomes operators achieved by using one or the other bioactive material in periodontal defects in general. One study was cited by the authors. Looking for defects with significantly diminished number of bone walls as this study did, there are no studies available by today comparing the efficacy of one vs. the other bioactive formulation. However, we have now included more literature to the discussion regarding EMD and xHyA use in other studies and also added Figure 5, which summarizes their unique mode of action.
- The manuscript does not adequately address potential biases in the study design, implementation, and analysis. A lack of consideration for biases could undermine the internal validity of the study and compromise the credibility of the results.
We agree with the reviewer’s assessment. However, there is an intrinsic risk of bias associated with the design of this study. A retrospective analysis of a case series, albeit reproting from one or from three centers, has limitations related to the study design. In this analysis, we tried to address these issues comprehensively by calibrating the surgical technique as well as the variables to be analyzed. Nevertheless, the obtained results are to be regarded with caution due to the lack of randomization, control groups or comparability of the applied heterografts. We believe that all these methodological concerns are addressed adequately throughout the (now revised) discussion and that another paragraph pointing to the inherent bias risks is unnecessary as these risks are inherent to the study design. In other words, the authors claim at no point, that this is a randomized controlled multicenter study.
Consider the following points to improve the quality of the manuscript:
- The term "biofunctionalized heterografts" is introduced but not explicitly defined. Consider providing a clear definition early in the introduction to aid reader comprehension.
Thank you for this comment. The following explanation was added to the M&M section:
„The concordant intention was to enhance tissue response to bone substitutes or membranes applied by combining them with bioactive formulations, either enamel matrix derivative (EMD) or high molecular weight crosslinked hyaluronic acid (xHyA) aiming at their biofunctionalization. “
- Specify the specific regenerative mechanisms targeted by enamel matrix derivative (EMD) and cross-linked heavy molecular weight hyaluronic acid (xHyA) in the context of periodontal regeneration.
As recommended by reviewer 1, we have added a section to the discussion and also provided a schematic illustration.
- Include studies that investigate the specific cellular and molecular mechanisms through which EMD and xHyA contribute to periodontal tissue regeneration.
Studies reflecting the molecular and cellular mechanisms of mode of action in periodontal regeneration are included.
- Clarify the rationale for selecting EMD and xHyA as biofunctionalized heterografts, highlighting their distinct properties and potential synergies.
Done.
- Provide more information on the standardization process for applying EMD and xHyA across different centers.
Each center was free to use the bioactive formulation according to its daily routine. As to EMD, the recommended original protocol was followed, i.e., the pre-conditioning using Pref-Gel for 2 minutes was carried out, followed by drying the defect and root surface before applying the EMD gel. The xHyA application neither requires specific pre-conditioning of the root and/or defect walls nor needs a dry bloodless condition. Therefore, this gel was directly applied to the defect and root surface once the mechanical instrumentation and degranulation were completed. The following passage was added to explain the application mode for both biomaterials.
“For the EMD application the site was pre-conditioned by using the 24% EDTA gel (Pref Gel, Straumann Group, IL) according to recommendation from the manufacturer taking care of bloodless condition of the defect area prior to EMD gel application thereafter. The xHyA application followed manufacturer’s recommendation also, however, any pre-conditioning of the site was redundant and therefore omitted. The re-hydration of either bone substitute was carried out on the tray before grafting the defect by particulate heterograft. The rehydration afforded as much bioactive material as necessary to completely cover the total volume of the graft. The amount of EMD used per site amounted in total to one dose of 0.7 ml while one ampule of xHyA contained 1.2 ml of the hyaluronic gel. … The membrane was re-hydrated by xHyA similarly to the re-hydration of bone substitute in the other two centers.”
- Clearly define the criteria used to classify a site as a success or failure, particularly in terms of pocket closure rates, to ensure consistency across centers.
This is mentioned in the Results section, we extended the statement by definition of criteria for pocket closure estimation as follows:
In detail, looking at residual PD ≤ 4mm without BoP, C1 showed 89.5%, C2 95.3% and C3 93.4% pocket closure rate at the level of residual probing depth of <4mm without bleeding on probing.
- Specify if there were any noticeable variations in the outcomes among different centers and discuss potential reasons for such variations.
We specified the changes between baseline and 12-months values by the Table 3 and discussed the differences in the Discussion section.
- Emphasize the specific advantages and disadvantages of using EMD and xHyA as biofunctionalized heterografts for periodontal regeneration.
These two items themselves represented just one part of a heterograft and were never combined. Nevertheless, we amended the discussion accordingly.
- Discuss the practical implications of the study's findings in the context of developing targeted regenerative therapies.
We extended the Discussion section to make it easier to understand which benefits both bioactive formulations offer for periodontal regeneration at a similar level of effectiveness.
- Summarize the unique contributions of the study to the understanding of periodontal regeneration using EMD and xHyA.
The Discussion is extended by a passage addressing this issue.
- Include figures or tables that visually compare the regenerative outcomes of EMD and xHyA across different centers.
Done. Please refer to Figure 1
- Ensure that recent and relevant studies exploring the regenerative potential of EMD and xHyA in periodontal tissues are included in the literature review.
Checked
- Discuss the potential translation of the study findings into clinical practice, addressing any challenges or considerations.
Considered
- Propose specific avenues for future research, such as investigating the long-term effects of EMD and xHyA in diverse patient populations.
Done
Round 2
Reviewer 5 Report
Comments and Suggestions for Authors
The conclusion section is not clear. Rewrite it proper description and conclude the results neatly.
Author Response
Thank you for this comment. We adapted the conclusions by following paragraph (page 13, lines 352-359):
In summary, this retrospective analysis of biofunctionalized heterografts across three centers demonstrated significant improvements in pocket probing depth reduction and attachment level gains. Center 3 showcased noteworthy clinical attachment level improvements without using particulate bone substitutes, emphasizing the potential benefits of hyaluronic acid application in GTR. While acknowledging study limitations, the findings highlight the effectiveness of biologics like EMD and hyaluronic acid in periodontal regenerative surgery, prompting the need for future randomized controlled trials to optimize treatment combinations for enhanced patient outcomes.
Furthermore we changed the data availability statement:
The data that support the findings of this study are available upon reasonable request from the corresponding author, A.F. The data are not publicly available due to ethical concerns regarding patient information.